# Extracellular Vesicles Derived from Osteogenic-Differentiated Human Bone Marrow-Derived Mesenchymal Cells Rescue Osteogenic Ability of Bone Marrow-Derived Mesenchymal Cells Impaired by Hypoxia

**DOI:** 10.3390/biomedicines11102804

**Published:** 2023-10-16

**Authors:** Chenglong Wang, Sabine Stöckl, Girish Pattappa, Daniela Schulz, Korbinian Hofmann, Jovana Ilic, Yvonne Reinders, Richard J. Bauer, Albert Sickmann, Susanne Grässel

**Affiliations:** 1Department of Orthopedic Surgery, Experimental Orthopedics, Center for Medical Biotechnology (ZMB), Biopark 1, University of Regensburg, 93053 Regensburg, Germanykorbinian.hofmann@stud.uni-regensburg.de (K.H.); 2Department of Trauma Surgery, Center for Medical Biotechnology (ZMB), Biopark 1, University of Regensburg, 93053 Regensburg, Germany; 3Department of Oral and Maxillofacial Surgery, Center for Medical Biotechnology (ZMB), Biopark 1, University Hospital Regensburg, 93053 Regensburg, Germanyrichard.bauer@ukr.de (R.J.B.); 4IZKF Group Tissue Regeneration in Musculoskeletal Diseases, University Hospital & Bernhard-Heine-Centrum for Locomotion Research, University of Würzburg, 97070 Würzburg, Germany; j-ilic.klh@uni-wuerzburg.de; 5Leibniz-Institut für Analytische Wissenschaften—ISAS—e.V., Bunsen-Kirchhoff-Straße 11, 44139 Dortmund, Germany; yvonne.reinders@isas.de (Y.R.); albert.sickmann@isas.de (A.S.); 6Medizinisches Proteom-Center, Ruhr-Universität Bochum, 44801 Bochum, Germany; 7Department of Chemistry, College of Physical Sciences, University of Aberdeen, Aberdeen AB24 3FX, UK

**Keywords:** extracellular vesicles, BMSC, osteogenic differentiation, hypoxia, normoxia, ROS, antioxidants

## Abstract

In orthopedics, musculoskeletal disorders, i.e., non-union of bone fractures or osteoporosis, can have common histories and symptoms related to pathological hypoxic conditions induced by aging, trauma or metabolic disorders. Here, we observed that hypoxic conditions (2% O_2_) suppressed the osteogenic differentiation of human bone marrow-derived mesenchymal cells (hBMSC) in vitro and simultaneously increased reactive oxygen species (ROS) production. We assumed that cellular origin and cargo of extracellular vesicles (EVs) affect the osteogenic differentiation capacity of hBMSCs cultured under different oxygen pressures. Proteomic analysis revealed that EVs isolated from osteogenic differentiated hBMSC cultured under hypoxia (hypo-osteo EVs) or under normoxia (norm-osteo EVs) contained distinct protein profiles. Extracellular matrix (ECM) components, antioxidants and pro-osteogenic proteins were decreased in hypo-osteo EVs. The proteomic analysis in our previous study revealed that under normoxic culture conditions, pro-osteogenic proteins and ECM components have higher concentrations in norm-osteo EVs than in EVs derived from naïve hBMSCs (norm-naïve EVs). When selected for further analysis, five anti-hypoxic proteins were significantly upregulated (response to hypoxia) in norm-osteo EVs. Three of them are characterized as antioxidant proteins. We performed qRT-PCR to verify the corresponding gene expression levels in the norm-osteo EVs’ and norm-naïve EVs’ parent cells cultured under normoxia. Moreover, we observed that norm-osteo EVs rescued the osteogenic ability of naïve hBMSCs cultured under hypoxia and reduced hypoxia-induced elevation of ROS production in osteogenic differentiated hBMSCs, presumably by inducing expression of anti-hypoxic/ antioxidant and pro-osteogenic genes.

## 1. Introduction

Poor blood circulation affects tissue oxygen saturation levels in the human body, creating hypoxic pathological conditions that are the underlying cause of many human diseases, such as brain ischemia, heart attack and acute lung and kidney injury [1,2]. In orthopedics, many diseases have common pathological conditions related to hypoxia. For example, femoral head osteonecrosis can develop due to blood supply disruption, which results in hypoxic injury to the femoral head. The bone mineral loss, which arises from arteriosclerosis of the lower limbs, together with hypoxic ischemia, can induce osteoporosis [3]. In addition, basic reports [4,5] support the theory that these orthopedic diseases’ pathological conditions are related to hypoxia. Hypoxia is able to impair bone regeneration by reducing the differentiation capacity of bone marrow-derived mesenchymal cells (BMSC) toward osteoblasts [6]. However, there are also other reports showing conflicting and inconsistent results regarding the influence of hypoxia on the osteogenic differentiation of precursor cells. Wagegg et al. [7] showed that osteogenic differentiation of naïve human (h)BMSCs is enhanced under hypoxic conditions compared to normoxic conditions. They concluded that hypoxia promotes osteogenesis of hBMSCs in a hypoxia-inducible factor (HIF)-1-dependent manner. HIFs are proteins that respond to changes in oxygen concentration and are subjected to proteosomal degradation under normoxia while they are stabilized under hypoxia [8]. 

‘Reactive oxygen species’ (ROS) is an umbrella term for an array of derivatives of molecular oxygen, including hydrogen peroxide (H_2_O_2_) and the superoxide anion radical (O_2_**^−^**). Changes in ROS production were shown to modulate diverse physiological processes [9,10], as excessive production of ROS is suggested to inhibit bone formation [11,12]. In general, hypoxic conditions affect ROS levels [13,14], suggesting that the ROS level is increased under hypoxia. Contrary, reports from other studies [15,16] showed that hypoxia caused a ROS level decrease compared to normoxic conditions in a pre-osteoblastic cell line and in macrophages. Thus, our **first aim** was to determine how hypoxia would affect the osteogenic differentiation ability of naïve hBMSCs and if it increases ROS production. 

Extracellular vesicles (EVs) are enclosed by a lipid bilayer and released by a wide range of cell types (under physiological and pathological conditions), with diameters ranging from 50 to 200 nm [17,18]. EVs carry a wide range of bioactive molecules, including proteins, lipids, mRNAs, microRNAs (miRNAs), and long noncoding RNAs (lncRNAs), with EVs’ cargo being directly dependent on the metabolic condition or the differentiation status of their parental cells [19,20,21]. Moreover, EVs, which function as cell-to-cell communicators, have emerged as an important route for interchanging proteins, lipids and genetic material between cells and tissues, similar to neurotransmitters acting as chemical messengers transmitting information between neurons [1,21]. In previous years, many therapeutic approaches for musculoskeletal disorders have been focused on MSC-based cell therapies due to their differentiation and immunomodulatory properties. MSC-EVs may account for a large part of these functions and are, therefore, under investigation as an alternative therapeutic approach to cell therapies. Immigrating MSCs into the bone defect secrete large amounts of EVs, and it was shown that MSC-EV-based approaches have the potential to promote bone regeneration [19].

EVs may play a pivotal role in cancer cell growth, progression, and metastasis in solid tumors [22]. It is well established that cell proliferation is rapid in tumors, but the vasculature formed is insufficient to maintain a sufficient oxygen level, leading to the development of a hypoxic tumor microenvironment [23,24]. The alteration in the composition and function of tumor-derived EVs mediated by hypoxia allows the tumor cells to respond to hypoxia and to modify their surrounding microenvironment [22,24]. Furthermore, several in vitro studies [25,26] suggested that EVs derived from tumor cells under hypoxia contain a unique protein signature that significantly enhances invasion compared to EVs from tumor cells grown under normoxia. However, according to our current knowledge, none of those studies addressed the analysis of EVs from osteogenic differentiated BMSCs under hypoxia. Therefore, the **second aim** of this study was to analyze the proteomes of EVs derived from osteogenic differentiated hBMSCs under normoxic or hypoxic conditions and to investigate whether osteogenic EVs generated under hypoxia are loaded with unique proteomic signatures, which would allow the cells to respond to hypoxia.

EVs are envisioned as promising bioactive effectors to promote the osteogenic differentiation capacity of MSCs, which induce the efficient repair of bone injuries [27,28]. Further, MSC-EVs can promote new bone formation with supporting vascularization and displaying improved morphological, biomechanical, and histological outcomes, coupled with positive effects on cell survival, proliferation, migration, osteogenesis, and angiogenesis [19]. 

Our previous study [29] showed that under normoxic conditions, EVs, isolated from hBMSCs at a late-stage of osteogenic differentiation, promote the osteogenic differentiation capacity of hBMSCs and, at the same time, have negative effects on the adipogenic differentiation capacity of hBMSCs. Current knowledge of the underlying molecular mechanisms of these pro-osteogenic and anti-adipogenic effects of EVs under hypoxia is very limited. Thus, in this study, the **third aim** was to further investigate why and how EVs derived from hBMSCs at late-stage osteogenic differentiation under normoxia enhance the osteogenic capacity of naïve hBMSCs cultured under hypoxia. The three aims of this study are summarized in Figure 1.

## 2. Materials and Methods

A graphical illustration of the experimental set-up is shown in Figure 1. 

### 2.1. Ethical Statement 

This study has been approved by the local ethics committee (MSCs: Ethikkommission, No. 14-101-0189, University of Regensburg), and all patients’ written informed consent was obtained before undergoing surgery (email: ethikkommission@klinik.uni-regensburg.de). 

### 2.2. Isolation, Culture and Characterization of Immunophenotype of Human BMSCs 

hBMSCs were prepared from frozen stocks obtained from the femoral bone marrow aspirate of twelve patients undergoing hip replacement surgery (mean age: 60.4 ± 8.7 years, range: 46–73 years, male: 50%). Density gradient centrifugation was used to isolate the hBMSCs according to established protocols [30,31,32]. hBMSCs were expanded until 80% confluency (passage 1–4) in StemMACS^TM^ MSC expansion medium (Miltenyi Biotec, Bergisch Gladbach, Germany) with the following composition: L-glutamine, fetal calf serum (FCS) (Sigma-Aldrich GmbH, Bergisch Gladbach, Germany), phenol red and supplemented with 0.2% MycoZap (Lonza, Basel, Switzerland). Flow cytometric analysis of hBMSC with specific antibodies against MSC positive marker (CD44, CD73) and negative marker (CD19, CD34) is shown in Appendix A.

### 2.3. Generation of EVs-Depleted FCS 

EVs-depleted FCS (FCS^depl-uc^) was prepared according to established protocols [33,34]. In short, FCS was diluted in α-MEM medium to a final concentration of 20% and subsequently ultra-centrifuged at 120,000× *g* (Beckman Coulter, Brea, CA, USA) overnight (18 h) at 4 °C to generate EV-depleted FCS. FCS^depl-uc^ was stored in aliquots at −20 °C. Medium used for EV collection and hBMSC stimulation was supplemented with 10% FCS^depl-uc^. 

### 2.4. Osteogenic Differentiation of Human BMSCs 

For harvesting the conditioned medium (CM) for EV isolation, 3 × 10^6^ hBMSCs (passage 4) were cultured in growth medium in triple T175 flasks (ThermoFisher, Dreieich, Germany) until 80% confluency before onset of osteogenic differentiation. To induce osteogenic differentiation of hBMSCs according to established protocols [32,34], the expansion medium was changed to osteogenic differentiation medium (α-MEM (Sigma-Aldrich, Steinheim, Germany), 10% FCS^depl-uc^ or 10% regular FCS, 4 mM GlutaMAXTM-I (Gibco, Paisley, UK), 1% penicillin–streptomycin (P/S), 10 μM ascorbic acid-2-phosphate, 10 mM ß-glycerophosphate and 100 nM dexamethasone (all from Sigma-Aldrich, Steinheim, Germany). Osteogenic differentiation was terminated after a maximum of 35 days (medium was replaced every 3 days). 

### 2.5. Induction of Hypoxia 

A total of 2% oxygen is defined as a hypoxic condition according to published protocols [35,36]. For osteogenic differentiation under hypoxia, hBMSCs were cultured in a hypoxia incubator (ThermoFisher Scientific, Langenselbold, Germany) set at 2% O_2_ and 5% CO_2_ levels, balanced with 93% N_2_. Normoxic control BMSCs were incubated in a standard cell culture incubator set at 20% oxygen, 5% CO_2_ and 75% N_2_. 

### 2.6. Collection of Conditioned Medium for EV Isolation

Worth noting, 3 × 10^6^ hBMSCs (passage 4) were expanded in triple T175 flasks in growth medium until 80% confluency under normoxia. 

For harvesting the norm-naïve EVs, conditioned medium (CM) for EV isolation from undifferentiated naïve hBMSCs (i.e., norm-naïve EVs = naïve hBMSCs derived EVs) under normoxia, growth medium was replaced by α-MEM medium with 1% P/S, 10% FCS^depl-uc^ and 4 mM GlutaMAXTM-I. After 48 h incubation under normoxic conditions, CM was stored immediately at −80 °C for subsequent EV isolation. 

Osteogenic differentiation of hBMSCs was induced under normoxic or hypoxic conditions, respectively. For harvesting osteogenic CM under normoxia and hypoxia, regular FCS was replaced by 10% FCS^depl-uc^ in the osteogenic medium on days 26, 28, 30 and 33 of culture. After 48 h incubation, CM was collected on days 28, 30, 32 and 35 under normoxic and hypoxic conditions, respectively. (i.e., hypo-osteo EVs containing CM = harvesting and pooling of CM from hBMSCs undergoing osteogenic differentiation from 28–35 days under hypoxia; norm-osteo EVs containing CM = harvesting and pooling CM after 28–35 days under normoxia) and stored at −80 °C for further processing.

### 2.7. EV Isolation

To isolate hypo-osteo EVs, norm-naïve EVs and norm-osteo EVs from the respective CM, ultracentrifugation was performed according to our previously published protocol [34,37]. In brief, the respective CM was centrifuged at 300× *g* (Sigma, Osterodea, Harz, Germany) for 10 min at 4 °C to remove intact cells. The supernatant was transferred to a new falcon tube and centrifuged again at 2000× *g* to remove dead cells. Afterward, the supernatant was transferred to a new tube and centrifuged for 30 min at 10,000× *g* to remove cell debris, and the subsequent supernatant was filtered (Filtropur S plus 0.2 μm; Sarstedt, Nümbrecht, Germany) into an ultracentrifugation tube (Polypropylene Centrifuge Tubes; Beckman Coulter, Brea, CA, USA) and was subjected to ultracentrifugation once at 120,000× *g* for 70 min at 4 °C. Following careful aspiration of the supernatant, the pellet was washed with PBS and centrifuged again (120,000× *g* for 70 min at 4 °C). The resulting EV pellets were resuspended in the presence of 25 mM trehalose (Carl Roth, Karlsruhe, Germany) in PBS. Protein concentration was measured using BCA Protein Assay Kit Pierce (Thermo Scientific, Rockford, IL, USA). 

### 2.8. Conditioned Medium pH under Hypoxia and Normoxia

CM for isolation of hypo-osteo EVs and norm-osteo EVs was obtained as described above. Osteogenic medium with 10% FCS^depl-uc^, kept in T75 flasks for two days either under hypoxia or normoxia (without cells), was set as control groups. Subsequently, all CM groups were collected, and pH of each CM was determined using a pH meter (Hanna Instruments, Nusfalau, Romania).

### 2.9. Subcellular Protein Extraction

Nuclear protein extracts were obtained using the NE-PER™ Nuclear and Cytoplasmic Extraction Kit and Halt™ Protease Inhibitor Cocktail according to the manufacturer’s instructions. The nuclear protein isolates were determined using a BCA Protein Assay Kit (all reagents are from Thermo Scientific, Rockford, IL, USA).

### 2.10. Western Blotting Analysis

#### 2.10.1. Detection of EV Markers 

From each sample, 10 μg purified EVs were loaded onto and separated by 15% SDS-PAGE. After electrophoretic separation, proteins were blotted onto 0.22 μm PVDF membranes (Carl Roche, Karlsruhe, Germany). The PVDF membranes were stained with Ponceau S staining solution (Sigma-Aldrich, Steinheim, Germany) and documented by photography for blotting efficiency and total protein expression analysis. Subsequently, the membranes were washed and blocked with 5% BSA (Carl Roth, Karlsruhe, Germany) in 0.1% Tween Tris Buffered Saline (T-TBS) for 1 h at RT and consecutively incubated with the following primary antibodies in 5% BSA/T-TBS overnight at 4 °C: anti-CD9 (1:1000), anti-CD63 (1:500) and anti-CD81 (1:1000) antibodies (all Thermo Fisher Scientific, Rockford, IL, USA) overnight at 4 °C. After three washing steps, 10 min each, membranes were incubated with horseradish peroxidase coupled secondary antibody (1:10,000) (Jackson Immuno Research, West Grove, PA, USA) for 1 h at RT. Protein bands were visualized with ECL detection reagents and SuperSignal™ West Femto Maximum Sensitivity Substrate (both Thermo Fisher Scientific, Rockford, IL, USA). Semi-quantitative analysis was performed with ImageJ 1.54 software (Bio-Rad, Hercules, CA, USA). 

#### 2.10.2. Detection of HIF-1α and RUNX2

Then, 3 × 10^5^ hBMSCs were cultured in growth medium (T75) until 80% confluency. Growth medium was exchanged with osteogenic differentiation medium containing 10% FCS under either normoxia or hypoxia, and cells were cultured for 14 days. Nuclear proteins were isolated using the NE-PER™ Nuclear and Cytoplasmic Extraction Kit. From each sample, 30 μg of nuclear protein was loaded onto and separated by 4–15% Mini-PROTEAN^®^ TGX™ Precast Protein Gels (Bio-Rad, Hercules, CA, USA). Western blot was performed as described above using anti-HIF-1α (1:1000), anti-RUNX2 (1:1000) and anti-Histone H3 (1:1000) antibodies (all Cell Signaling Technology, Danvers, MA, USA).

### 2.11. EV Uptake Test

Similar volumes of hypo-osteo EVs (15 μg) and norm-osteo EVs (15 μg) were labeled with PKH26 Red Fluorescent Cell Linker Mini Kit for general cell membrane labeling (Sigma-Aldrich, Saint Louis, MO, USA) according to our previous publication [29]. Subsequently, 2 × 10^4^ naïve hBMSCs were cultured in growth medium in eight-well chamber slides (Falcon, Big Flats, NY, USA) for 48 h. Cells were washed with PBS, and the pre-stained EVs (15 μg) were added for 24 h. Nuclei of cells were counterstained with DAPI (Molecular Probes, Eugene, OR, USA) and the cytoskeleton with Phalloidin (Abcam, Malvern, UK), then staining was analyzed using a fluorescence microscope (Eclipse TE2000-U; Nikon, Tokyo, Japan).

### 2.12. Nanoparticle Tracking Analysis (NTA)

The concentration and particle size distribution of the purified EV fractions were analyzed using The NanoSight NS300 (Malvern Instruments, Malvern, UK) following the manufacturer’s instructions. Briefly, the accuracy of NTA was confirmed with 100 nm polystyrene beads (Sigma-Aldrich, Saint Louis, MO, USA), then EV samples were diluted 1:100 in PBS at RT and a total of five 30 s videos were recorded.

### 2.13. Alkaline Phosphatase Assay

Intracellular alkaline phosphatase (ALP) enzyme activity was determined with QuantiChrom™ Alkaline Phosphatase Assay Kit (BioAssay Systems, Hayward, CA, USA). Moreover, 1 × 10^4^ hBMSCs were cultured in 24-well plates in growth medium under normoxia until 80% confluency. 

Then, growth medium was replaced by osteogenic differentiation medium with 10% FCS under either normoxia or hypoxia (2% O_2_), and cell culture was continued for two weeks. For EV treatment, hBMSCs were cultured under hypoxia for two weeks in osteogenic differentiation medium containing 10% FCS^depl-uc^ and were treated with the different EV groups (10 μg/mL) or PBS (no EVs) from days 8 to 14 (fresh EVs were added every two days). hBMSCs kept in osteogenic differentiation medium containing 10% FCS^depl-uc^ under normoxia for two weeks were set as positive control group (no EVs). Afterward, intracellular ALP enzyme activity was quantified in all groups. Data were calculated either as percentage of the ‘no EVs‘ group under hypoxia (negative control group) or as percentage of normoxia group (no EV treatment = positive control).

### 2.14. Alizarin Red Staining

For analyzing calcification levels, 3 × 10^4^ hBMSCs were cultured in 12-well plates in growth medium until 80% confluency. For comparison of hypoxic and normoxic conditions, growth medium was replaced by osteogenic differentiation medium containing 10% FCS under either normoxia or hypoxia for three weeks. 

For EV treatment, cells were cultured under hypoxia with osteogenic differentiation medium containing 10% FCS^depl-uc^ for 3 weeks and stimulated with the different EV groups (10 μg/mL) or PBS (no EVs) from days 5 to 21 (fresh EVs and PBS were added every two days). Cells, which were kept in osteogenic differentiation medium containing 10% FCS^depl-uc^ under normoxia for three weeks, were set as positive control group (no EVs). Subsequently, cells were washed and fixed with glutaraldehyde for 15 min at RT. After a washing step with PBS (pH = 4.2), cells were incubated for 20 min with Alizarin Red-S staining solution (1%, Carl Roth, Karlsruhe, Germany) at RT. Afterward, quantitative analysis was performed as described previously [34]. Results were further calculated as percentage of the ‘no EVs group’ under hypoxia (negative control group).

### 2.15. RNA Isolation and Real-Time -qPCR

For analyzing gene expression, 6 × 10^4^ hBMSCs were cultured in growth medium (6-well plates) until 80% confluency. For comparison of hypoxic and normoxic conditions, growth medium was exchanged for osteogenic differentiation medium containing 10% FCS under either normoxia or hypoxia for two weeks. For EV treatment, growth medium was exchanged to osteogenic differentiation medium with 10% FCS^depl-uc^, and cells were cultured for 2 weeks under hypoxia and treated with the different EV groups (10 μg/mL) or PBS (no EVs) from day 12 on. Cells kept in osteogenic differentiation medium containing 10% FCS^depl-uc^ under normoxia for two weeks were set as positive control group (no EVs). Afterward, RNA was isolated using Absolutely RNA™ Microprep Kit (Agilent Technologies, Cedar Creek, TX, USA), and cDNA was prepared using AffinityScript QPCR cDNA Synthesis Kit (Agilent Technologies, Cedar Creek, TX, USA) as recommended by manufacturer’s protocol. RT-qPCR was performed in duplicates using the Brilliant III Ultra-Fast SYBR Green QPCR Master Mix with an Agilent PCR-System (Agilent Technologies, Cedar Creek, TX, USA). All genes were analyzed relatively in relation to GAPDH and TATA-binding protein (TBP) expression (normalizer). All genes were set to the expression of the calibrator (‘no EVs’ group under hypoxia).

For validation of proteomics data, norm-naïve EVs parent cells (undifferentiated, naïve hBMSCs) were harvested for RNA isolation under normoxia. For harvesting RNA of the norm-osteo EVs’ parent cells, the growth medium was replaced by osteogenic medium with 10% FCS^depl-uc^ and cells were cultured for either 14 or 35 days under normoxia. All genes were analyzed and calibrated to the expression in naïve hBMSCs under normoxia (calibrator). Primers are listed in Appendix A.

### 2.16. Measurement of Reactive Oxygen Species (ROS) Level 

Extracellular ROS level was measured with the OxiSelect™ in vitro ROS/RNS Assay Kit (Cell Biolabs Inc., San Diego, CA, USA). 1 × 10^4^ hBMSCs were cultured in 24-well plates in growth medium under normoxia until 80% confluency. Growth medium was replaced by osteogenic differentiation medium, which was changed every third day. 

For comparison of hypoxic and normoxic conditions, hBMSCs were cultured in osteogenic differentiation medium containing 10% FCS for either 14 or 35 days under either normoxia or hypoxia. Osteogenic differentiation medium was changed every third day. After 48 h incubation at day 12 and day 33, the osteogenic supernatants were collected subsequently on days 14 and 35.

For EV treatment, hBMSCs were cultured under hypoxia for two weeks in osteogenic differentiation medium containing 10% FCS^depl-uc^. The different EV groups (10 μg/mL) or PBS (no EVs) were added from days 12 to 14. hBMSCs, which were kept in osteogenic differentiation medium containing 10% FCS^depl-uc^ under normoxia for two weeks, were set as positive control group (no EVs). Results were further calculated as percentage of the ‘no EVs group’ under hypoxia (negative control group).

Extracellular ROS was quantified according to manufacturer’s instructions. Briefly, 50 μL of cell supernatants were added into a 96-well black-bottom fluorescence plate (NunclonTM, Thermo Fisher Scientific, Roskilde, Denmark), and 50 μL of catalyst were added to each well and incubated for 5 min at RT. Afterward, 100 μL of DCFH was added at RT for 15 min in the dark. Fluorescence was measured using a SpectraMax^®^ iD3 plate reader (Molecular Devices, San Jose, CA, USA) at 480 nm excitation/530 nm emission. Fluorescence intensity is proportional to the total ROS levels within the sample.

### 2.17. Proteomic Analysis 

#### 2.17.1. Sample Preparation of EVs for Proteomics Analysis

EV samples were purified from the culture supernatant of the corresponding three cell donors, and total protein concentration was measured using the BCA assay. Briefly, 5 μg EV protein was reduced in 10 mM dithiothreitol (DTT) at 56 °C for 30 min and consecutively alkylated in 30 mM of iodoacetamide (IAA) at RT for 30 min in the dark. The remaining IAA was quenched with 30 mM DTT at RT for 15 min. Afterward, EV proteins were digested using the S-Trap^TM^ (ProtiFi, Fairport, NY, USA) mini procedure according to the manufacturer’s protocol. 

#### 2.17.2. Quantitative Proteomic Analysis by LC-MS/MS 

From each EV sample, 500 ng per protein was analyzed by nano LC-MS/MS. EV samples were loaded on an Ultimate 3000 Rapid Separation Liquid chromatography (RSLC) nano system with a ProFlow flow control device coupled to a Lumos Fusion orbitrap mass spectrometer (both from Thermo Scientific). Peptides dissolved in 0.1% TFA were placed onto a trapping column (Acclaim PepMap100 C18, 100 μm × 2 cm, Thermo Scientific, Bremen, Germany) at a flow rate of 10 μL/min. Afterward, peptides were separated on a phase column (Acclaim PepMap100 C18, 75 μm × 50 cm, Thermo Scientific, Bremen, Germany) using a binary gradient.

#### 2.17.3. Database Search and Bioinformatics Analysis 

All MS raw data were identified using the Proteome Discoverer software 2.3.0.523 (Thermo Scientific, Bremen, Germany), and it is MASCOT algorithm against a human UniprotKB database (http://www.uniprot.org; downloaded 21 November 2019). The search parameters were 0.5 Da for MS and MS/MS and precursor and fragment ion tolerances of 10 ppm, respectively. Carbamidomethylation of cysteine was set as fixed modification, and oxidation of methionine was set as dynamic modification. Trypsin was set as enzyme with a maximum of two missed cleavages, using Percolator false discovery rate (strict) set to 0.01 for both peptide and protein identification. Label-free quantification (LFQ) analysis was performed, including replicates for each condition. Proteins with more than 2-fold change were considered as distinct proteins. All distinct upregulated and downregulated proteins in hypo-osteo EVs compared with those in norm-osteo EVs were then subjected to Kyoto Encyclopedia of Genes and Genomes (KEGG) and Gene Ontology (GO) analyses. The following bioinformatic analyses include Venn diagram, heatmap, KEGG and GO databases. Among them, Venn diagram was conducted with the bioinformatics platform jvenn [38] (http://jvenn.toulouse.inra.fr/app/index.html, accessed on 21 November 2019), and the heatmap was constructed with the software TBtools 1.0 [39]. GO and KEGG analysis of annotated proteins were conducted with the bioinformatics platform STRING database (http://string-db.org, accessed on 21 November 2019) and visualized using following bioinformatics platform (https://www.bioinformatics.com.cn, accessed on 21 November 2019). In order to obtain a better idea of the potential relationships between the proteins, protein–protein interaction networks of the identified proteins were constructed with the STRING database with default parameters and visualized using Cytoscape software 3.9.1.

### 2.18. Statistical Analysis

Prism 8.21 software (GraphPad, San Jose, CA, USA) was used for statistical analysis. Differences between groups were assessed by One sample t-test and Wilcoxon test or by one-way ANOVA with Dunn’s multiple comparisons test when appropriate. *p* < 0.05 was considered statistically significant. 

## 3. Results

### 3.1. Generating Conditioned Medium for Preparation of Osteogenic EVs from a Hypoxic (Hypo-Osteo EVs) Environment and Osteogenic EVs from a Normoxic (Norm-Osteo EVs) Environment

Representative images in Appendix A show that the color of conditioned medium (CM) from osteogenic differentiated hBMSCs under hypoxic conditions is distinct from CM under normoxic conditions and also distinct from osteogenic medium (OM) (no cells) subjected to both hypoxic and normoxic conditions (Appendix A). In order to explain the different medium colors under normoxic and hypoxic conditions, the pH-values of CM and OM were determined. As shown in Appendix A, the pH of CM from osteogenic differentiated hBMSCs under hypoxia decreased significantly compared to CM recovered from osteogenic differentiated hBMSCs under normoxia and from both OM subjected to both hypoxia and normoxia.

### 3.2. Characterization of EVs

#### 3.2.1. NTA Evaluation of EVs

We determined the particle concentration, distribution and size of hypo-osteo EVs and norm-osteo EVs via nanoparticle tracking analysis (NTA). The average particle size of hypo-osteo EVs (n = 3) was smaller by trend than their norm-osteo-EV counterparts, although both representatively shown EV groups correspond to the standard size of EVs (Figure 2A,B,D). However, there is no statistically significant difference in counts and size between hypo-osteo EVs and norm-osteo EVs (Figure 2C,D).

#### 3.2.2. Uptake of EVs by Naïve hBMSCs

In order to evaluate the cellular internalization of EVs, naïve hBMSCs were incubated with PKH-26 stained hypo-osteo EVs and norm-osteo EVs for 24 h. Intracellular fluorescence labeling (red) revealed that the internalized hypo-osteo EVs and norm-osteo EVs accumulated in the cytoplasm (Figure 2E,F “merge”) of the target hBMSCs with no obvious differences between both groups.

#### 3.2.3. Surface Markers of EVs

The presence of the most common canonical EV membrane markers CD9, CD63 and CD81 was analyzed via Western blotting. Positive bands for CD9, CD63 and CD81 were detected in both hypo-osteo-EV and norm-osteo-EV groups (Figure 2G). Appendix A show the respective uncropped Western blot membranes after development and the respective blot membranes stained with Ponceau S as loading control. The quantification of each marker band intensity showed that CD9, CD63 and CD81 protein expression levels were significantly decreased in hypo-osteo EVs compared to norm-osteo EVs (Figure 2H–J). Quantitative proteomic analyses confirmed the presence of CD9, CD63 and CD81 in the EV samples (Figure 2K). Uncropped Western blot membranes and Ponceau S stained loading control images are shown in Appendix A.

### 3.3. Osteogenic Differentiation of hBMSCs under Hypoxia

To assess the influence of hypoxia on the matrix calcification ability of osteogenic differentiated hBMSCs, analysis of calcified matrix nodules after 21 days of osteogenic differentiation under normoxia and hypoxia was conducted via Alizarin Red staining. Figure 3A demonstrates that calcium deposits in the hypoxia sample groups decreased in intensity and area compared to the normoxia samples, even though the staining intensity showed inter-sample differences. In general, the quantification of the Alizarin Red staining intensity and amount confirmed that the BMSCs of all five donors cultured under hypoxia have significantly decreased calcium deposits compared to those cultured under normoxia after 21 days of osteogenic differentiation. Results from different donors were combined for quantification (Figure 3B). Alkaline phosphatase (ALP) enzyme activity level serves as an indicator of bone formation and correlates with osteoblast activity [40]. Here, the ALP activity assay revealed that compared to the normoxia group, hypoxia significantly suppresses ALP activity in hBMSCs after 14 days of osteogenic differentiation (Figure 3C). To assess whether hypoxia influences the ROS level of osteogenic differentiated hBMSCs, we performed a ROS assay after 14 and 35 days of osteogenic differentiation under hypoxic and normoxic conditions. Appendix A shows that the extracellular ROS level was already significantly increased after 14 days of osteogenic differentiation of hBMSCs under hypoxic conditions compared to normoxic conditions. Moreover, excessive production of ROS was maintained up to 35 days of osteogenic differentiation under hypoxia (Appendix A).

We further analyzed gene expression levels of the osteogenic markers BGLAP (Osteocalcin), RUNX2 (Runt-related transcription factor 2), ALP, COL1A1 (Collagen alpha-1(I) chain) and OPN (Osteopontin). Besides OPN, the expression of all analyzed genes revealed a significant decrease under hypoxia compared to normoxia (Appendix A).

### 3.4. HIF-1α and RUNX2 Protein Expression in hBMSCs Undergoing Osteogenic Differentiation under Hypoxia

In order to evaluate if a 2% O_2_ condition is effective for induction of hypoxia in cells, we analyzed hypoxia-inducible factor-1α (HIF-1α) protein expression, and if hypoxia influences osteogenic differentiation of hBMSCs, we analyzed RUNX2 protein expression. Western blotting analysis showed that hypoxia induces HIF-1α protein expression and downregulates RUNX2 protein expression in hBMSCs undergoing osteogenic differentiation. Nuclear protein Histone H3 served as loading control. Results from four different donors were combined for quantification (Figure 3D–F). Uncropped Western blot membranes and Ponceau S-stained loading control images are shown in Appendix A.

### 3.5. Proteomic Analysis of Osteogenic EVs Produced under Hypoxia and Normoxia

#### 3.5.1. Summary of the Proteomic Profiles 

To investigate whether osteogenic EVs from a hypoxic environment are loaded with unique proteomic signatures, we performed quantitative proteomic analysis to identify the protein profiles of hypo-osteo EVs (n = 3) and norm-osteo EVs (n = 3). The Venn diagram in Figure 4A shows that 52 unique proteins were identified in the norm-osteo EV group, and 11 unique proteins were identified in the hypo-osteo EV group. Moreover, the norm-osteo EV group has 913 proteins in common with the hypo-osteo EV group. Analysis of distinct proteins with at least a 2-fold change between the hypo-osteo EV group and the norm-osteo EV group revealed that the protein levels of 67 proteins were increased and the levels of 377 proteins were decreased in the hypo-osteo EVs compared to the norm-osteo EVs (Appendix A). As shown in Figure 4B, the detailed list of the distinct proteins (more than a 2-fold change) and their abundance is presented in the heatmap. We identified 21 pro-osteogenic proteins, 21 proteins involved in ECM remodeling and five antioxidant proteins, which were decreased in hypo-osteo EVs compared to that in norm-osteo EVs (Table 1, Table 2 and Table 3). The hypo-osteo EVs/norm-osteo EVs ratios measured for these proteins ranged from a 100-fold to a 2008-fold reduction. The five decreased antioxidant proteins are: 5’-AMP-activated protein kinase catalytic subunit alpha-1 (PRKAA1), NAD(P)H dehydrogenase 1 (NQO1), Annexin A1 (ANXA1), 40S ribosomal protein S3 and Peroxiredoxin-1 (RPS3) and Peroxiredoxin-1 (PRDX1). 

#### 3.5.2. GO and KEGG Enrichment Analyses 

GO (Gene Ontology) terms were performed for annotation of the differentially expressed proteins (more than 2-fold change) in norm-osteo EVs and hypo-osteo EVs, which are divided into three categories (biological processes, molecular functions and cell components). GO biological processes analysis of upregulated and downregulated proteins in hypo-osteo EVs were compared with that in norm-osteo EVs. Among them, upregulated proteins matched 42 significant GO terms (no GO terms related to bone regeneration). The top 19 upregulated Go terms are shown in Figure 5A. Downregulated proteins matched 650 significant GO terms, and the significant downregulated GO terms related to bone regeneration (ECM, osteogenesis, ROS, adhesion and angiogenesis) are also shown in Figure 5A. GO analysis revealed information about all upregulated and downregulated molecular functions, specifically concerning the downregulated cell-to-cell and cell-to-matrix binding processes (Figure 5B) and cellular components (Figure 5C) in hypo-osteo EVs compared to norm-osteo EVs. KEGG pathway analysis identified significantly downregulated and upregulated enriched pathways in hypo-osteo EVs compared to that in norm-osteo EVs (Figure 5D). The pathways, shown in Figure 5D, revealed that especially the ‘PI3K-Akt signaling’, ‘ECM-receptor interaction’ ‘cGMP-PKG signaling pathway’ and ‘growth hormone synthesis, secretion and action’, were significantly downregulated in hypo-osteo EVs compared with norm-osteo EVs.

#### 3.5.3. Hub Proteins

Protein–protein interaction (PPI) networks of all the differently expressed proteins (more than a 2-fold change) were created using the STRING database and visualized using Cytoscape with unconnected nodes excluded. Altogether, 520 nodes (62 upregulated and 358 downregulated proteins in hypo-osteo EVs compared to that in norm-osteo EVs) and 2680 interaction pairs were identified in the PPI network of the distinct proteins (Figure 6A). Hub nodes contain the key genes of the PPI network. The top 20 hub proteins of the whole PPI network were identified using the Degree and CytoHubba plug-ins in Cytoscape (Figure 6B). Among them, the two most highly correlated hub proteins are epidermal growth factor receptor (EGFR) and catenin beta-1 (CTNNB1), which are downregulated in hypo-osteo EVs compared to norm-osteo EVs. Notably, EGFR was identified in the context of GO terms ‘cellular response to reactive oxygen species’, pro-osteogenic and pro-angiogenic proteins’. Moreover, CTNNB1 was identified in the context of pro-osteogenic and pro-angiogenic proteins (Table 1). As shown in Figure 6B, the two hub proteins are marked with a yellow box.

### 3.6. Effects of EVs Derived from Late Stage Osteogenic-Differentiated hBMSCs under Normoxia on Osteogenic Ability of hBMSCs Cultured under Hypoxia

To investigate the capability of norm-osteo EVs to influence matrix calcification of osteogenic differentiated hBMSCs impaired by hypoxia, Alizarin Red staining of calcified matrix was evaluated after 21 days of osteogenic differentiation under hypoxia and simultaneous EV stimulation (EVs from osteogenic differentiated and naïve hBMSCs cultured under normoxia) for the last 16 culture days. Representative images showing calcium deposits revealed that norm-osteo EVs induced increased intensity and area of calcification compared to the negative control group (no EVs under hypoxia) and compared to EVs isolated from undifferentiated naïve hBMSCs (norm-naïve EVs) when added to hBMSCs (Figure 7A). The quantification of the Alizarin Red staining intensity was consistent with these results. The norm-osteo EV group significantly increased calcium deposits compared to the negative control group (no EVs under hypoxia) and the norm-naïve EV group when added to hBMSCs; however, the area of calcium deposits was still smaller compared to the positive culture group (no EVs under normoxia) (Figure 7B). 14 days of osteogenic differentiation of hBMSCs in the presence of norm-naïve EVs and norm-osteo EVs (applied for the last six culture days), resulted in increased ALP activity compared to the negative control group (Figure 7C). In addition, the ‘no EV’ treatment group cultured under normoxia still displayed a higher ALP activity level in comparison to stimulation with norm-naïve EVs and norm-osteo EVs under hypoxia. 

To investigate the capability of norm-osteo EVs to reduce elevated ROS levels in osteogenic differentiated hBMSCs under hypoxia, ROS concentration was determined after 14 days of osteogenic differentiation under hypoxia and simultaneous EV stimulation. Figure 8D showed that the norm-osteo EV group and norm-naïve EV group significantly reduced ROS levels compared to the negative control group (no EVs under hypoxia). Furthermore, there was no significant difference in ROS production among the two EV groups and the positive control group.

Analysis of osteogenic markers revealed that treatment with the norm-osteo EVs significantly increased RUNX2 expression in hBMSCs compared to the norm-naïve EVs and no EVs control groups under hypoxia (Figure 7E). No significant difference was found in ALP, OPN, BGLAP and COL1A1 gene expression between hBMSCs stimulated with hypo-naïve and hypo-osteo EVs (Figure 7F–I), whereas ALP and BGLAP gene expression in the positive control group (no EVs under normoxia) was significantly higher.

#### 3.6.1. GO Enrichment Analyses and Protein–Protein Interaction (PPI) Network Analyses of Norm-Naïve EVs and norm-Osteo EVs

Our previous study [29] revealed that norm-osteo EVs and norm-naïve EVs contained distinct protein profiles, with pro-osteogenic proteins and ECM components highly enriched in norm-osteo EVs. In order to determine the potential molecules that mediate the effects of osteogenic EVs in rescuing the osteogenic ability of hBMSCs impaired by hypoxia, GO biological processes analysis of upregulated proteins in norm-osteo EVs compared to that in norm-naïve EVs were further analyzed, and the five significant GO terms related to hypoxia and osteogenesis are shown in Appendix A. To further investigate the potential molecular factors contained in norm-osteo EVs that suppress hypoxia and promote osteogenesis, we performed Venn diagram (Appendix A) and PPI network analyses on upregulated proteins in norm-osteo EVs associated with ‘response to hypoxia’, ‘ossification’, ‘skeletal system development’, ‘osteoblast differentiation’ and ‘extracellular matrix (ECM) organization’ (Appendix A). Four protein interaction networks at significant levels (PPI enrichment *p* value < 0.05) and one protein interaction network (response to hypoxia) at no significant level (PPI enrichment *p* value = 0.102) were constructed. As shown in Appendix A, ‘response to hypoxia’ has three proteins (Caveolin-1 (CAV1)/Secreted frizzled-related protein 1 (SFRP1) and/or matrix metalloproteinase-2 (MMP-2) in common with the ‘ossification’, ‘skeletal system development’, ‘osteoblast differentiation’ or ‘ECM organization’. The analysis indicates a significant relationship between the CAV1/SFRP1/MMP-2 and the other proteins in the four GO terms related to osteogenesis and ECM (PPI enrichment *p* value < 0.05). 

#### 3.6.2. Hub Proteins

The Cytoscape platform confirmed 148 nodes (96 upregulated and 52 downregulated proteins in norm-osteo EVs compared to those in norm-naïve EVs), and 653 interaction pairs were identified in the PPI network of the distinct proteins (Appendix A). The top 20 hub proteins of the whole PPI network were identified using the Degree and CytoHubba plug-ins in Cytoscape (Appendix A). Among them, 11 hub proteins (Caveolin-1, Fibulin-1, Collagen alpha-1(VI) chain, Nidogen-2, Collagen alpha-3(VI) chain, 72 kDa type IV collagenase, Elastin, Decorin, Fibronectin, Basement membrane-specific heparan sulfate proteoglycan core protein, MAGUK p55 subfamily member 2) related to hypoxia or osteogenesis, were increased in the norm-osteo EVs group compared to the norm-naïve EVs group. 

#### 3.6.3. Anti-Hypoxic Proteins

As shown in Appendix A, a total of nine proteins were identified in the GO terms related to ‘response to hypoxia’, which are highly enriched in norm-osteo EVs compared with that in norm-naïve EVs, and they were selected for further analysis. Among them, five anti-hypoxic proteins were experimentally determined in the previous studies [41,42,43,44,45], which are Caveolin-1, Secreted frizzled-related protein 1, Extracellular superoxide dismutase (SOD), Endoplasmin (i.e., 94 kDa glucose-regulated protein or HSP90B) and Aquaporin-1 (AQP1). Three of these five proteins (Caveolin-1, Extracellular superoxide dismutase-3 and Endoplasmin) are also classified as antioxidant proteins [46,47,48]. In this line, the norm-osteo EVs/norm-naïve EVs ratios measured for these five proteins ranged from a 5.09-fold to a 100-fold increase (Figure 8A).

### 3.7. Validation of Proteomics Data 

#### 3.7.1. Anti-Hypoxic Gene Expression Levels in the EVs’ Parent Cells

To assess the reliability of the proteomic data from norm-osteo EVs and norm-naïve EVs, we selected the five anti-hypoxic genes (CAV1, SFRP1, SOD3, HSP90B1 and AQP1) and further performed qRT-PCR to verify their relative expression levels in the norm-osteo EVs’ and norm-naïve EVs’ parent cells cultured under normoxia. We observed that CAV1, SFRP1, and AQP1 gene expression was significantly increased after 14 days (Appendix A) and after 35 days (Figure 8B–F) of osteogenic differentiation of hBMSC compared to naïve hBMSC both cultured under normoxia. 

#### 3.7.2. Anti-Hypoxic Gene Expression Levels in hBMSCs Undergoing Osteogenic Differentiation under Hypoxia after EV Treatment 

Under hypoxia, gene expression analysis of these five anti-hypoxic genes (CAV1, SFRP1, SOD3, HSP90B1 and AQP1) revealed that stimulation with both the norm-naïve EV and norm-osteo EV groups significantly increased CAV1 and HSP90B1 expression in hBMSCs compared to the negative control group (no EVs under hypoxia). Moreover, the SFRP1 gene expression level after norm-osteo EV stimulation was significantly increased compared to the negative control group. SOD3 gene expression level in hBMSCs after norm-naïve EVs stimulation was significantly increased compared to the negative control group (Figure 9A–E).

## 4. Discussion

We characterized EVs isolated from naïve or osteogenic differentiated hBMSCs cultured under normoxia and hypoxia and confirmed their purity and identity in line with our previous studies [42,43,46]. Hypoxia can suppress the osteogenic differentiation capacity of naïve hBMSCs, correlated to decreased calcium deposits, reduced ALP activity and downregulation of gene/protein expression of the osteogenic markers BGLAP, RUNX2, ALP, and COL1A1. This confirms an earlier study by Pattappa et al. [49], who showed that hypoxia may enhance long-term MSC expansion and reduce cell senescence but results in a population with impaired osteogenic differentiation potential. Our proteomic data supported these findings, revealing that hypoxia affected EVs’ protein cargo. Moreover, 21 pro-osteogenic proteins were reduced in hypo-osteo EVs compared to norm-osteo EVs (Table 1). In particular, Golgi apparatus protein 1 (GLG1), Transmembrane protein 119 (TMEM119) and alkaline phosphatase, tissue-nonspecific isozyme (ALPL) concentration were reduced about 6.5-fold to 100-fold in hypo-osteo EVs compared to norm-osteo EVs. GLG1 (alternative name: E-selectin ligand 1 = ESL-1) was recently described as an important regulator of bone remodeling, and loss of GLG1 in osteoblasts is leading to delayed differentiation and mineralization [50]. TMEM119 (alternative name: Osteoblast induction factor = OBIF) is an osteoblast differentiation factor that plays a vital role in bone formation and osteoblast differentiation [51], and ALPL (i.e., ALP) as an indicator of bone formation can reflect the activity of the osteoblasts [30]. Here, we clearly demonstrated that a hypoxic culture environment reduced the osteogenic activity of hBMSCs-derived EVs as it significantly suppressed ALP enzyme activity, expression of osteogenic genes, and ALP protein amount in hypo-osteo EVs’ cargo. Furthermore, the top two hub proteins, EGFR and CTNNB1, which were less concentrated in hypo-osteo EVs, are pro-osteogenic proteins. EGFR is critically important for osteogenesis, as EGFR deficiency leads to irregular mineralization of bone in mice [52]. CTNNB1 can upregulate osteogenic differentiation and mineralization via FOXQ1, which also promotes the nuclear translocation of CTNNB1 in murine (m)BMSCs, enhancing Wnt/β-catenin signaling, which was also shown to be essential for the osteogenic differentiation-promoting effect of FOXQ1 in mBMSCs [53]. In bone, the ECM is an intricate dynamic structure that plays an important role by instructing the osteogenic differentiation process of BMSCs [54]. The proteomic profiling of EVs showed that the concentration of 21 ECM proteins was decreased in hypo-osteo EVs compared to norm-osteo EVs (Table 2), and subsequent KEGG pathways analysis confirmed that ECM-receptor interaction pathways were significantly reduced in hypo-osteo EVs compared with norm-osteo EVs. ECM macromolecules promote the osteogenic differentiation process of BMSCs through ECM–receptor interaction, e.g., via integrin receptors. Integrins are heterodimeric cell surface receptors that mediate cell–ECM contact and subsequent adhesion and are critical in signal transduction from the ECM into the cells and vice versa [55,56]. CD9, CD81 and CD63 members of the tetraspan superfamily are associated with integrins and form different complexes with different members of the integrin family [57,58,59,60]. CD81 and CD9 identified as hub proteins in the PPI network, were decreased in hypo-osteo EVs compared to norm-osteo EVs, as validated with proteomic profiling and Western blot analysis. We assume that CD81 and CD9 are associated with osteogenesis, which is consistent with the study of Singh et al. [61], showing that osteo-chondro-progenitor cells expressing CD9 were readily differentiated to osteoblasts compared to cells without CD9, both in vitro and in vivo. 

Cellular oxidative stress may be defined as an imbalance between free radicals (i.e., ROS) and antioxidant proteins in favor of free radicals, potentially leading to oxidative damage of cell membranes and intracellular organelles [62,63]. In our study, hypoxia increased ROS production and secretion. Supported by our proteomic data, we demonstrated that hypoxia reduced the amount of several antioxidant proteins in the EVs’ cargo. Five antioxidants (5’-AMP-activated protein kinase catalytic subunit alpha-1 = PRKAA1, NAD(P)H quinone dehydrogenase 1 = NQO1, Annexin A1 = ANXA1, 40S ribosomal protein S3 = RPS3, Peroxiredoxin-1 = PRDX1) were clearly reduced in hypo-osteo EVs compared to norm-osteo EVs (Table 3). Zhu et al. [64] reported that PRKAA1 inactivation led to enhanced mitochondrial ROS production. NQO1, a superoxide scavenger, assists cells in handling ROS [65] and ANXA1, as an anti-inflammatory agent, can interfere with the generation of ROS [66]. RPS3 exerts antioxidative functions and protects cells against oxidative stress [67]. PRDX1 is described as a scavenger of ROS, and loss of PRDX1 in mice leads to an elevation of ROS. Thus, the proteomic analysis of hypo-osteo EVs provided novel insights into the putative mechanisms by which hypoxia can increase ROS production because antioxidant proteins were less abundant in hypo-osteo EVs, disturbing the subtle balance between extracellular ROS level and the intercellular transported antioxidants inside the EVs.

EVs derived from distinct cell types preferring a hypoxic environment, such as cancer cells, MSCs and cardiac cells, significantly increased angiogenesis compared to their normoxic EV controls [1,68]. Notably, our proteomic data revealed that the GO terms related to angiogenesis were significantly downregulated in hypo-osteo EVs derived from osteogenic differentiated BMSCs under hypoxia compared to the norm-osteo EV counterparts. Moreover, the top hub protein in our proteomic analysis is EGFR, a pro-angiogenic protein, which was less abundant in hypo-osteo EVs compared to norm-osteo EVs. It is known that osteonecrosis and osteoporosis pathogenesis are promoted by blood supply disruption, which results in hypoxic injury in bone. Thus, we speculate that under hypoxic pathological conditions in bone tissue, the alteration in the composition/cargo and function of osteogenic EVs, mediated by hypoxia, affects bone homeostasis, which in turn affects osteogenesis and angiogenesis negatively. Moreover, in our study, pro-angiogenic proteins were enriched in norm-osteo EVs, consistent with the study of Narayanan et al. [69], showing that norm-osteo EVs have a strong capacity for promoting vascularization in vivo. 

We clearly demonstrated that norm-osteo EVs, harvested at a late stage of osteogenic differentiation of hBMSCs under normoxia, can rescue the osteogenic ability of hBMSCs impaired by hypoxia. This was demonstrated by increased calcium deposits, enhanced ALP activity and the induction of gene expression of the osteogenic transcription factor RUNX2. Our previous study [29] revealed that under normoxic culture conditions, pro-osteogenic proteins were enriched in norm-osteo EVs in comparison to norm-naïve EVs from undifferentiated hBMSCs. Here, we further suggest that the five anti-hypoxic proteins (Caveolin-1 = CAV1, Secreted frizzled-related protein 1 = SFRP1, Extracellular superoxide dismutase-3 = SOD3, Endoplasmin = HSP90B1 and Aquaporin-1 = AQP1), which were highly enriched in norm-osteo EVs, might prevent cell damage or mitigated altered response to external stimuli inflicted by hypoxia. Caveolin-1 can reduce HIF1-α transcriptional activity under hypoxia by reducing HIF-1α S-nitrosylation in vitro and in vivo [35]. For secreted frizzled-related protein 1, it was recently reported that it can directly protect cells from apoptosis during hypoxia and reoxygenation [32]. Overexpression of SOD3 (i.e., EcSOD) suppressed the hypoxia-induced accumulation of HIF-1α in cells. When neurons were exposed to hypoxia/reoxygenation, cells with overexpression of HSP90B1 (i.e., GRP94) were resistant to apoptosis induced by hypoxia/reoxygenation [31]. Aquaporin 1 is a water and oxygen channel that can suppress hyperglycemia-induced cellular hypoxia [34]. The gene expression levels of CAV1, SFRP1 and AQP1 analyzed in the EVs’ parent cells (naïve and osteogenic differentiated hBMSCs kept under normoxia) followed the same trend as those indicated by the proteomic data of norm-osteo EVs and norm-naïve EVs supporting our proteomic data. In addition, after stimulation of hBMSCs kept under hypoxic conditions with EVs, we observed that norm-osteo EVs induced CAV1, SFRP1 and HSP90B gene expression in the osteogenic differentiated hBMSCs. Hence, we speculate that norm-osteo EVs are able to transfer CAV1, SFRP1 and HSP90B1 cargo to target cells in a hypoxic environment, and we hypothesize that the cargo of osteogenic EVs enable them to rescue the osteogenic ability of hBMSCs by transferring these and other proteins to the target cells. Moreover, our data suggest that norm-naïve EVs can also promote the anti-hypoxic capacity of a cell by the induction of the expression of CAV1, SOD3 and HSP90B1 in the target cells. That is a likely reason why norm-naïve EVs were able to promote ALP activity in hBMSCs cultured under hypoxia. 

Interestingly, three (CAV1, SOD3 and HSP90B1) of the five identified anti-hypoxic proteins are also known to act as antioxidants [36,37,38,70]. In our study, norm-osteo EVs could inhibit hypoxia-induced elevation of ROS produced by osteogenic differentiated hBMSCs, presumably by an increased antioxidant gene expression (CAV1 and HSP90B1). Moreover, norm-naïve EVs suppressed hypoxia-induced elevation of ROS levels in cultures of osteogenic differentiated hBMSCs, presumably by the increased antioxidant gene expression of SOD3, CAV1 and HSP90B1. Our results are consistent with the study of Khanh et al. [71], showing that EVs from infant MSCs rescued elderly MSCs from oxidative cell damage due to the elevation of ROS via upregulation of SOD1 and SOD3 protein expression. Furthermore, Ma et al. [14] reported that Icariin, a major bioactive pharmaceutical constituent isolated from Chinese medicine Horny Goat Weed, rescued the osteogenic ability of osteoblasts impaired by hypoxia via reducing the production of ROS, increasing SOD and ALP activity and forming a mineralized matrix. Thus, we speculate that norm-osteo EVs and norm-naïve EVs rescued the ALP activity of hBMSCs impaired by hypoxia by reducing the production of ROS. However, norm-naïve EVs were not able to rescue the mineralization ability of osteogenic differentiated hBMSCs impaired by hypoxia. According to our previous study [29], together with the current data presented here, a possible reason for that might be the fact that ECM components, pro-osteogenic proteins and anti-hypoxic proteins were less abundant in norm-naïve EVs compared to norm-osteo EVs.

## 5. Conclusions

In conclusion, the proteomic analysis of hypo-osteo EVs provides novel insights into how hypoxia can suppress the osteogenic ability of hBMSCs and simultaneously promote ROS production and secretion. The present study provides evidence that ECM macromolecules and -receptors, antioxidant- and pro-osteogenic proteins are decreased in hypo-osteo EVs, which otherwise would allow the osteogenic differentiated hBMSCs to respond to hypoxia and to accommodate to their surrounding microenvironment. These novel findings add to the understanding that the production of particular hypo-osteo EVs under hypoxic conditions contributes to impaired osteogenesis of mesenchymal precursor cells. Furthermore, norm-osteo EVs rescued the osteogenic ability of hBMSCs, impaired by hypoxia, by inducing expression of anti-hypoxic genes (CAV1, SFRP1, HSP90B1) and reversed hypoxia-induced elevation of ROS production in osteogenic differentiated hBMSCs. 

## Figures and Tables

**Figure 1 biomedicines-11-02804-f001:**
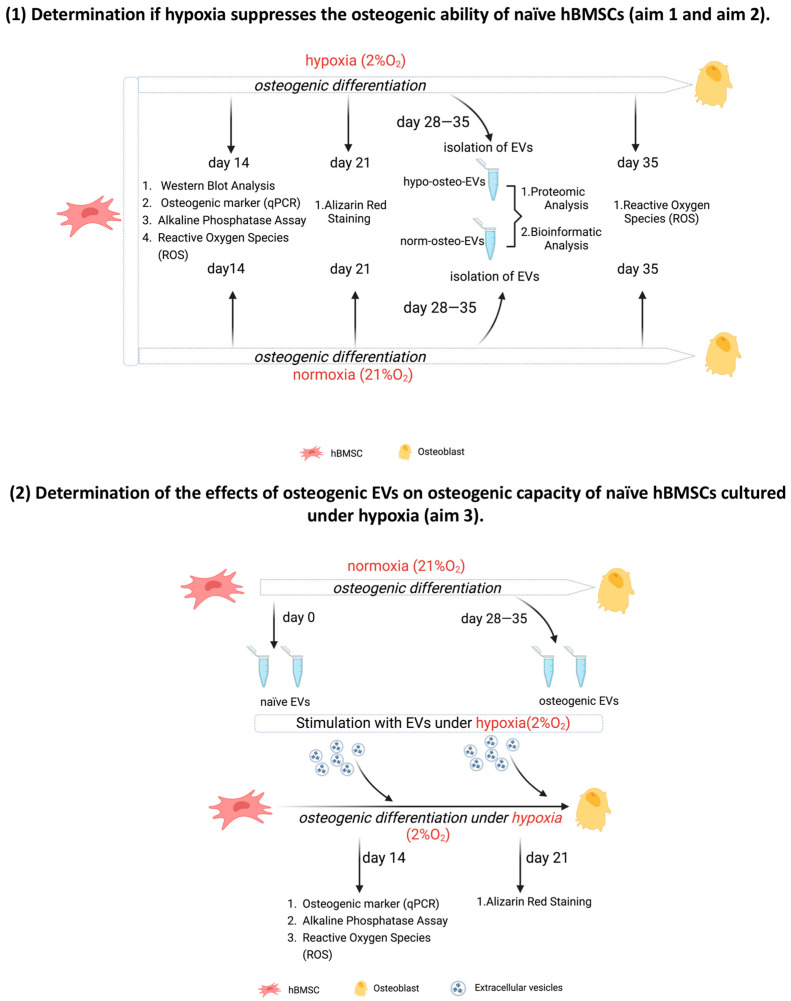
Overview of experimental set-up; EVs = extracellular vesicles.

**Figure 2 biomedicines-11-02804-f002:**
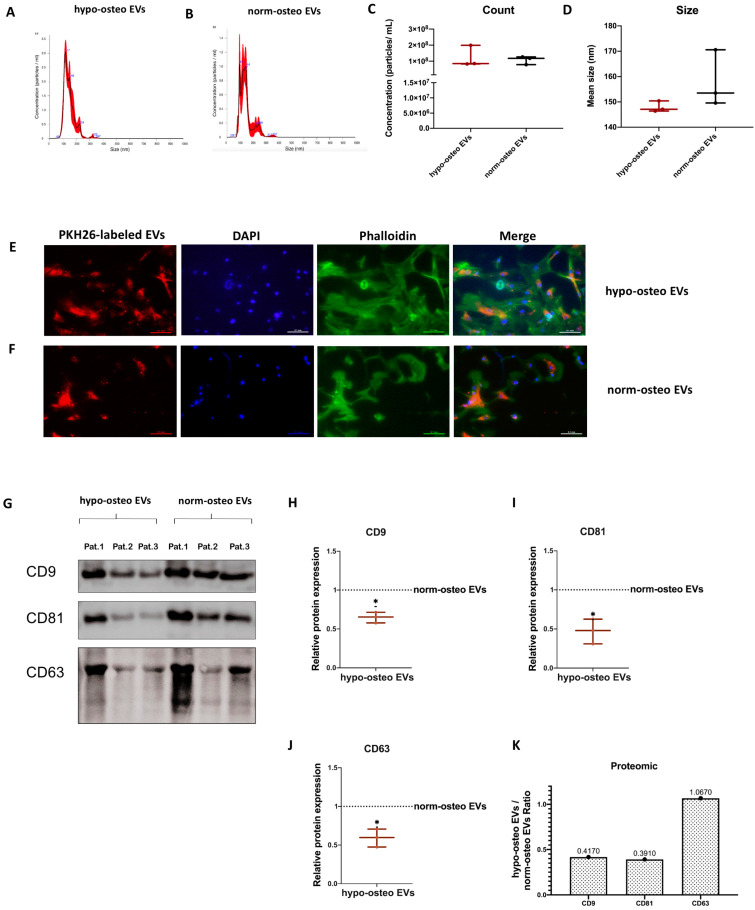
Characterization of the EV groups. (**A**,**B**) Representative particle size distribution of hypo-osteo EVs and norm-osteo EVs was measured by NTA; n = 3. (**C**,**D**) Quantitative comparison between hypo-osteo EVs and norm-osteo EVs in count and size measured; n = 3. (**E**,**F**) Uptake of EVs by naïve hBMSCs. PKH26-labeled hypo-osteo EVs (**E**) and norm-osteo EVs (**F**) were internalized by naïve hBMSCs and visualized with fluorescence microscopy. Cell nuclei were stained with DAPI, and structure of the cytoskeleton was visualized with Phalloidin staining. 20 × 10 magnification; Scale bar 100 μm; n = 3. (**G**) Western blot image showing bands of standard surface markers (CD9, CD81 and CD63) of hypo-osteo EVs and norm-osteo EVs; n = 3; Pat. = patient. (**H**–**J**) Relative quantitation of western blot image band intensities; n = 3. (**K**) The expression/level ratio of CD9, CD81 and CD63 proteins in hypo-osteo EVs was compared with that in norm-osteo EVs (proteomics data); n = 3. Results were calculated as percentage of the unstimulated control group (norm-osteo EVs, shown by the dotted line); * = *p* < 0.05.

**Figure 3 biomedicines-11-02804-f003:**
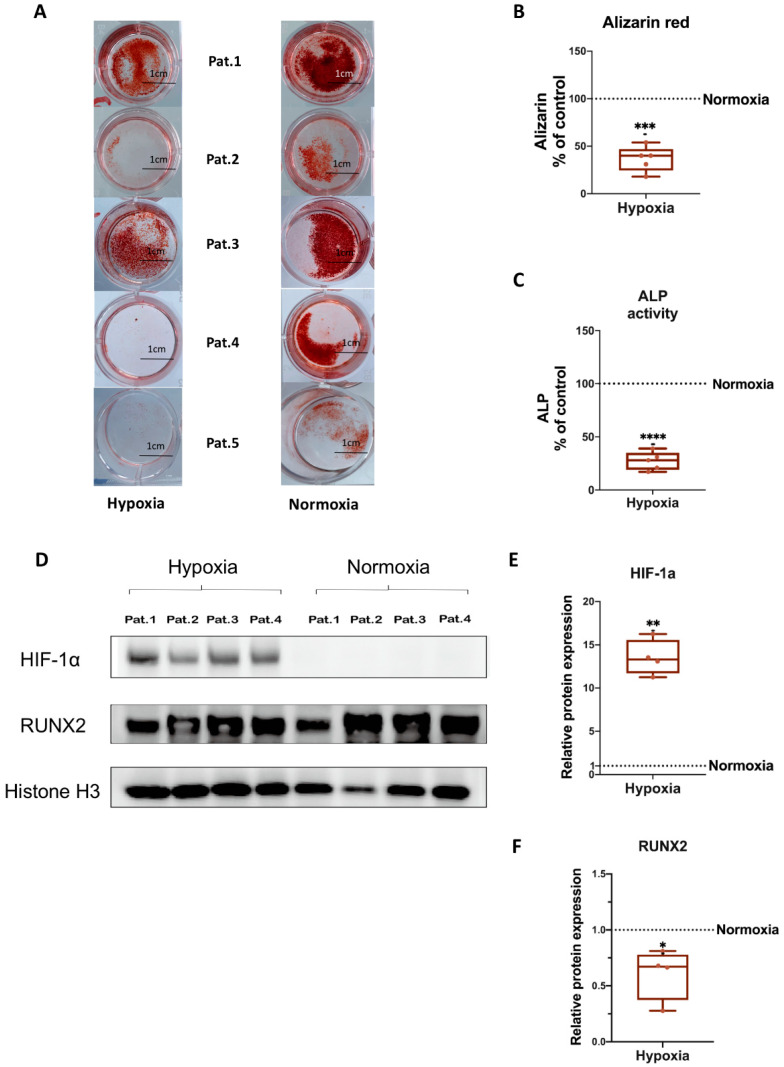
Evaluation of osteogenic differentiation ability of naïve hBMSCs under hypoxia and normoxia. (**A**) Alizarin Red staining of hBMSCs after 3 weeks of osteogenic differentiation under hypoxia or normoxia. Shown are the individual results from five different donors. Macroscopic view (scale bar 1 cm); n = 5; Pat. = patient. (**B**) Quantification of Alizarin Red staining; n = 5. (**C**) Quantification of Alkaline Phosphatase (ALP) activity of hBMSCs after 2 weeks of osteogenic differentiation under hypoxia or normoxia; n = 5. (**D**) Representative Western blot images of HIF-1α and RUNX2 after 2 weeks of osteogenic differentiation of hBMSCs under hypoxia and normoxia; n = 4. (**E**,**F**) Relative quantitation of Western blot image band intensities relative to Histone H3; n = 4. Results were calculated as percentage of the unstimulated control group (osteogenic differentiation of hBMSCs under normoxia, shown by the dotted line); * = *p* < 0.05; ** = *p* < 0.01; *** = *p* < 0.001; **** = *p* < 0.0001.

**Figure 4 biomedicines-11-02804-f004:**
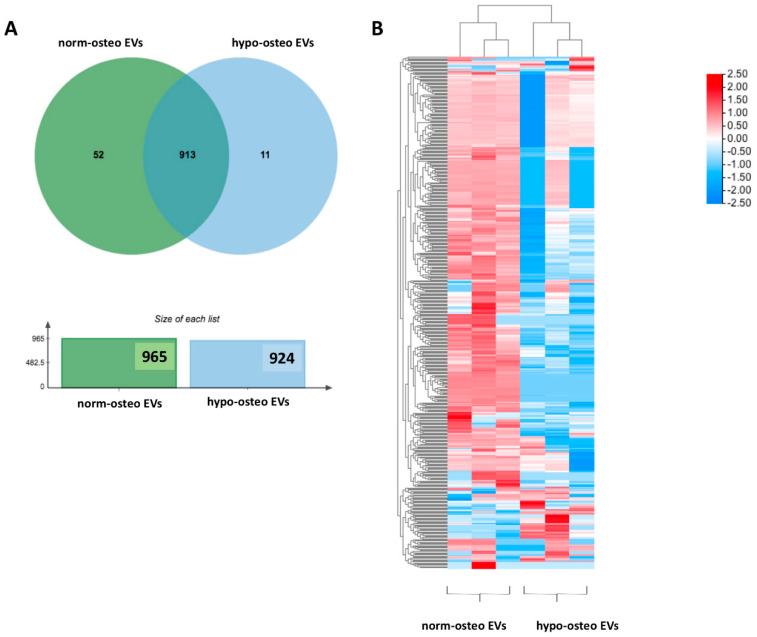
Venn diagram of total proteins and heatmap of distinct proteins identified in hypo-osteo EVs and norm-osteo EVs. (**A**) The distinct profiles (Venn diagram) of total proteins in hypo-osteo EVs and norm-osteo EVs; n = 3. (**B**) The distinct protein (more than 2-fold change) profiles of norm-osteo EVs and hypo-osteo EVs (heatmap); n = 3. The color code indicates the log2 (FC) difference of the proteins for those two EV groups: red means enriched in EVs, and blue means depleted in EVs.

**Figure 5 biomedicines-11-02804-f005:**
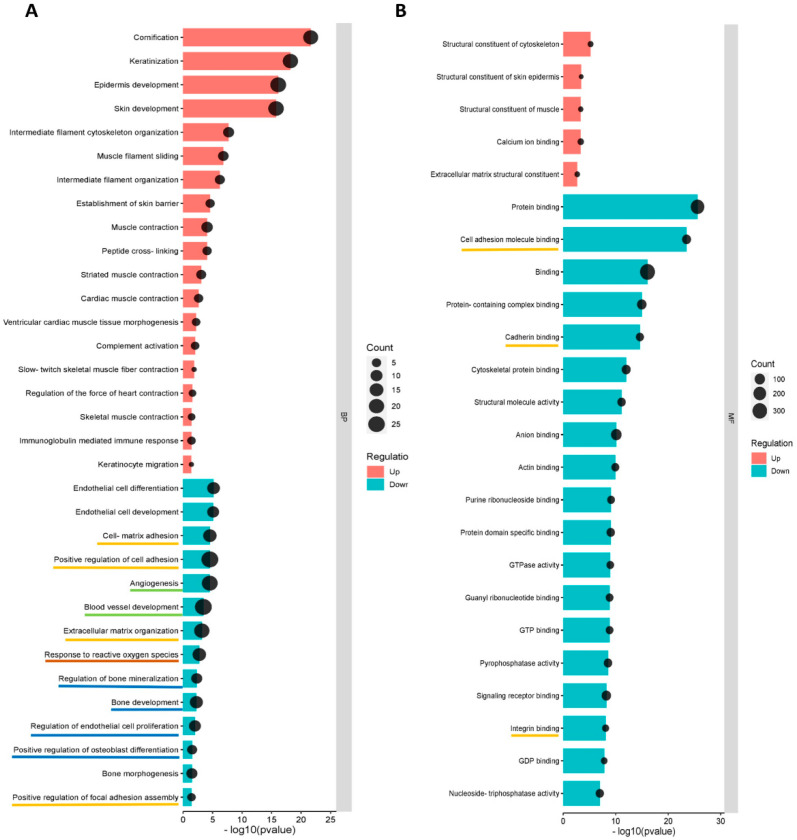
Functional enrichment analysis of distinct regulated and downregulated proteins in hypo-osteo EVs compared with that in norm-osteo EVs. Gene ontology (GO) analysis of the upregulated and downregulated biological processes (**A**), molecular functions (**B**), hypo-osteo EVs compared with that in norm-osteo EVs for upregulated and downregulated proteins were clustered; n = 3. (**C**) Cellular components in hypo-osteo EVs compared with that in norm-osteo EVs with the KEGG (**D**) enrichment analyses data for upregulated and downregulated proteins were clustered; n = 3. KEGG = Kyoto Encyclopedia of Genes and Genomes. Pathways related to bone regeneration are marked with pink lines. BP = biological processes; MF = molecular functions; CC = cellular components. GO terms related to bone regeneration (ECM, osteogenesis, angiogenesis, ROS and adhesion) and EV are marked with different colored lines.

**Figure 6 biomedicines-11-02804-f006:**
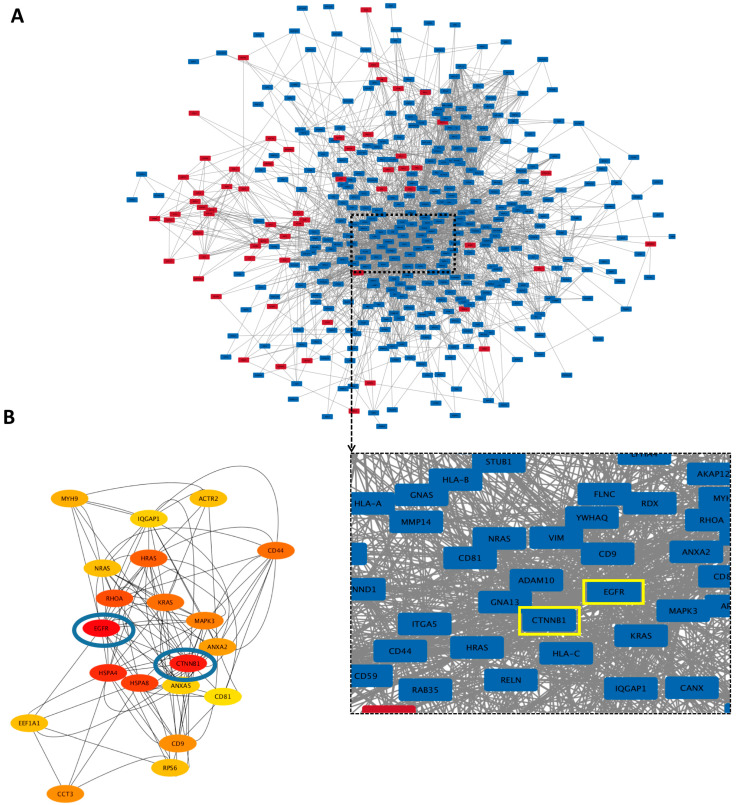
Protein–protein interaction (PPI) network of the identified proteins and hub proteins in hypo-osteo EVs compared with that in norm-osteo EVs. (**A**) Interactions between upregulated and downregulated proteins in hypo-osteo EVs compared with that in norm-osteo EVs; n = 3; Red nodes indicate upregulated proteins and blue nodes indicate downregulated proteins. (**B**) The 20 most highly correlated hub proteins in PPI network. The colors indicate the strength of correlated hub proteins of top 20 hub proteins; red is the highest correlated hub. The top two hub proteins (EGFR and CTNNB1) are marked with blue circles and yellow boxes.

**Figure 7 biomedicines-11-02804-f007:**
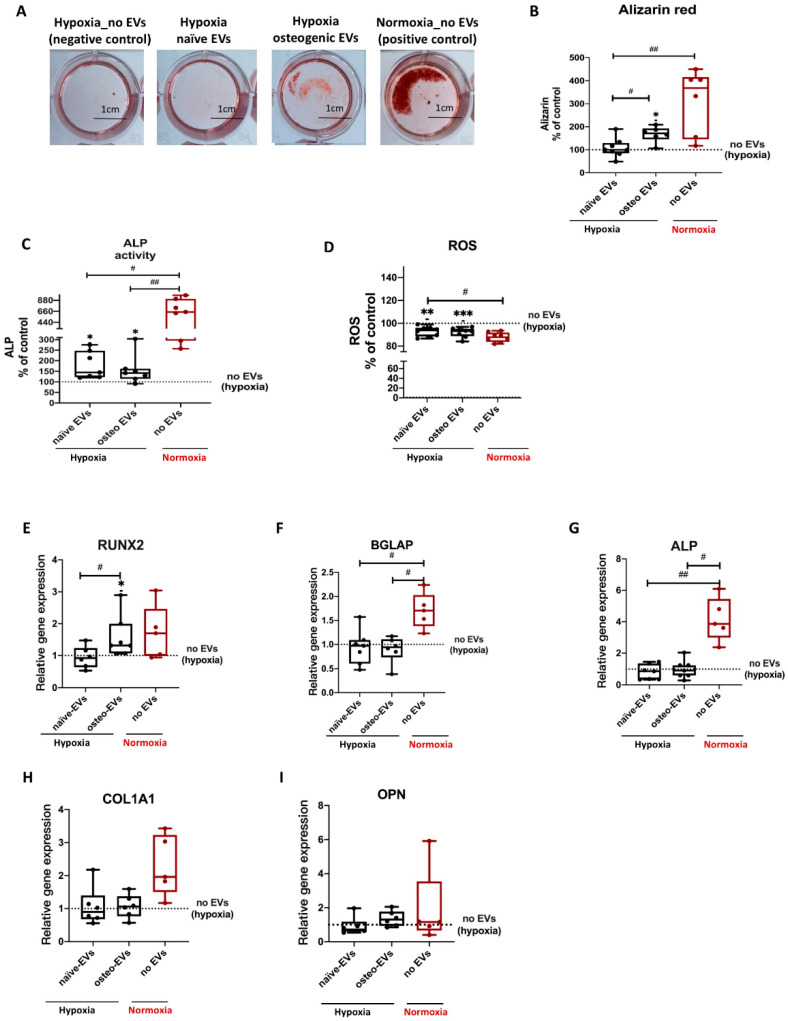
Evaluation of osteogenic differentiation ability of hBMSCs after EV treatment under hypoxia. (**A**) Alizarin Red staining of hBMSCs after 3 weeks of osteogenic differentiation and simultaneous stimulation with the different EV groups (scale bar 1 cm); n = 6–7. (**B**) Quantification of Alizarin Red staining; n = 6–7. (**C**) Quantification of ALP activity of hBMSC after 2 weeks of osteogenic differentiation and simultaneous stimulation with the different EV groups under hypoxia; n = 7. (**D**) ROS production of hBMSCs after 2 weeks of osteogenic differentiation and simultaneous stimulation with the different EV groups under hypoxia; n = 8. (**E**–**I**) Gene expression of the osteogenic marker genes (RUNX2, BGLAP, ALP, COL1A1 and OPN) were analyzed after 2 weeks of osteogenic differentiation of hBMSCs and simultaneous treatment with the different EV groups under hypoxia; n = 5–6. Results were calculated as percentage of the negative control group (no EVs under hypoxia, shown by the dotted line). Difference to the negative control: * = *p* < 0.05; ** = *p* < 0.01; *** = *p* < 0.001; difference between groups: ^#^= *p* < 0.05; ^##^ = *p* < 0.01.

**Figure 8 biomedicines-11-02804-f008:**
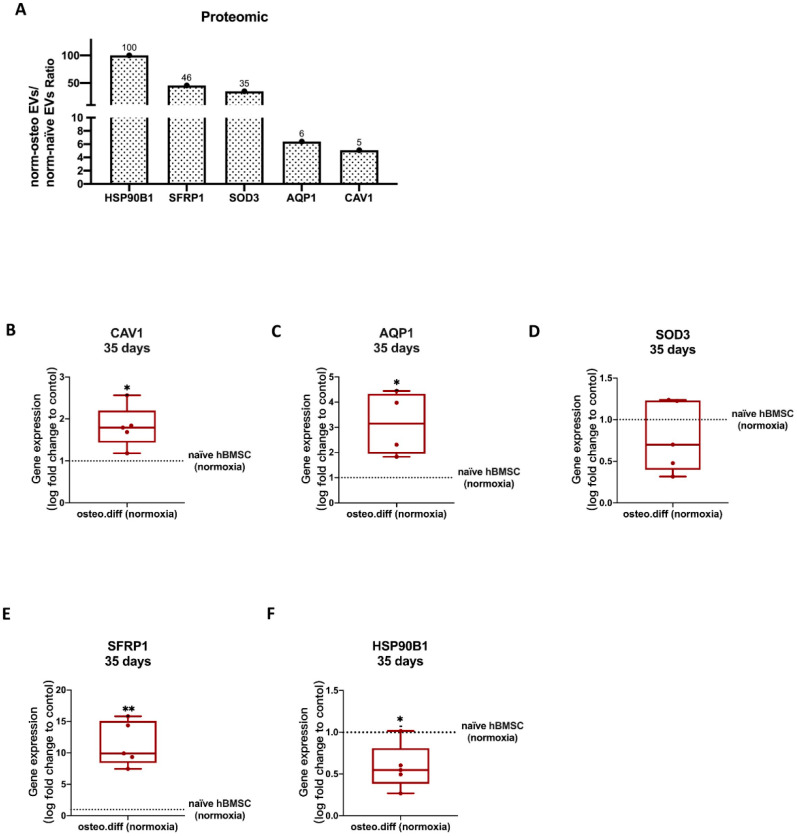
Validation of selected proteomic data obtained from norm-naïve EVs and norm-osteo EVs by gene expression analysis of hBMSCs. (**A**) The expression/level ratio of the five identified anti-hypoxic proteins (CAV1, SFRP1, SOD3, HSP90B1 and AQP1) in norm-osteo EVs compared with that in norm-naïve EVs (proteomics data); n = 3. (**B**–**F**) Gene expression of the five anti-hypoxic genes (CAV1, SFRP1, SOD3, HSP90B1 and AQP1) were analyzed in hBMSCs after 35 days of osteogenic differentiation and in naïve hBMSCs both cultured under normoxia; n = 4–5. Results were calculated as percentage of the control group (naïve hBMSCs under normoxia, shown by the dotted line); * = *p* < 0.05; ** = *p* < 0.01.

**Figure 9 biomedicines-11-02804-f009:**
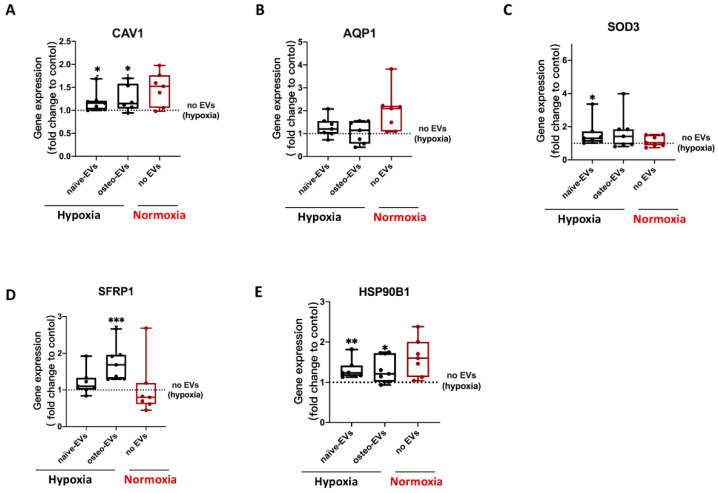
Evaluation of gene expression of the five anti-hypoxic proteins during osteogenic differentiation of hBMSCs under hypoxia after EVs stimulation. (**A**–**E**) Expression levels of the five anti-hypoxic genes were analyzed after 2 weeks of osteogenic differentiation in hBMSCs and simultaneous treatment with the different EV groups under hypoxia; n = 6–7. Results were calculated as percentage of the negative control group (no EVs under hypoxia, shown by the dotted line); Difference to the negative control: * = *p* < 0.05; ** = *p* < 0.01, ***= *p* < 0.001.

**Table 1 biomedicines-11-02804-t001:** The expression/level ratio of selected downregulated pro-osteogenic proteins in hypo-osteo EVs compared with those in norm-osteo EVs.

UniProt	Protein Names	Gene Names	Hypo-Osteo EVs/Norm-Osteo EVs Ratio (Fold Change)
Q92896	Golgi apparatus protein 1	GLG1	0.01
Q4V9L6	Transmembrane protein 119	TMEM119	0.106
P05186	Alkaline phosphatase	ALPL	0.154
P17813	Endoglin	ENG	0.245
Q04771	Activin receptor type-1	ACVR1	0.265
Q9ULC3	Ras-related protein Rab-23	RAB23	0.287
P26373	60S ribosomal protein L13	RPL13	0.292
P20020	Plasma membrane calcium-transporting ATPase 1	ATP2B1	0.303
P35222	**Catenin beta-1**	**CTNNB1**	0.304
P00533	**Epidermal growth factor receptor**	**EGFR**	0.305
Q16610	Extracellular matrix protein 1	ECM1	0.326
P50281	Matrix metalloproteinase-14	MMP14	0.331
P08133	Annexin A6	ANXA6	0.356
P07355	Annexin A2	ANXA2	0.362
P13797	Plastin-3	PLS3	0.387
Q16832	Discoidin domain-containing receptor 2	DDR2	0.401
P22413	Ectonucleotide pyrophosphatase/phosphodiesterase family member 1	ENPP1	0.438
Q5JWF2	Guanine nucleotide-binding protein G(s) subunit alpha isoforms XLas	GNAS	0.448
O00299	Chloride intracellular channel protein 1	CLIC1	0.458
Q13491	Neuronal membrane glycoprotein M6-b	GPM6B	0.494
P61586	Transforming protein RhoA	RHOA	0.498

**Table 2 biomedicines-11-02804-t002:** The expression/level ratio of selected downregulated ECM proteins in hypo-osteo EVs compared with that in norm-osteo EVs.

UniProt	Protein Names	Gene Names	Hypo-Osteo EVs /Norm-Osteo EVs Ratio (Fold Change)
Q12965	Unconventional myosin-Ie	MYO1E	0.01
P02458	Collagen alpha-1(II) chain	COL2A1	0.01
Q01955	Collagen alpha-3(IV) chain	COL4A3	0.01
Q12965	Unconventional myosin-Ie	MYO1E	0.01
Q16610	Extracellular matrix protein 1	ECM1	0.326
O75578	Integrin alpha-10	ITGA10	0.189
P17813	Endoglin	ENG	0.245
P26006	Integrin alpha-3	ITGA3	0.256
P06756	Integrin alpha-V	ITGAV	0.323
Q13683	Integrin alpha-7	ITGA7	0.326
P50281	Matrix metalloproteinase-14	MMP14	0.331
P18084	Integrin beta-5	ITGB5	0.347
P98095	Fibulin-2	FBLN2	0.347
Q08722	Leukocyte surface antigen CD47	CD47	0.352
P07355	Annexin A2	ANXA2	0.362
Q14112	Nidogen-2	NID2	0.379
P24821	Tenascin	TNC	0.396
Q16832	Discoidin domain-containing receptor 2	DDR2	0.401
P16070	CD44 antigen	CD44	0.412
O14672	Disintegrin and metalloproteinase domain-containing protein 10	ADAM10	0.428
P08648	Integrin alpha-5	ITGA5	0.498

**Table 3 biomedicines-11-02804-t003:** The expression/level ratio of selected downregulated antioxidant proteins in hypo-osteo EVs compared with that in norm-osteo EVs.

UniProt	Protein Names	Gene Names	Hypo-Osteo EVs /Norm-Osteo EVs Ratio (Fold Change)
Q13131	5’-AMP-activated protein kinase catalytic subunit alpha-1	PRKAA1	0.143
P15559	NAD(P)H dehydrogenase [quinone] 1	NQO1	0.298
P04083	Annexin A1	ANXA1	0.343
P23396	40S ribosomal protein S3	RPS3	0.438
Q06830	Peroxiredoxin-1	PRDX1	0.485

## Data Availability

Not applicable.

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
