# Peer review of "Extracellular Vesicles Derived from Osteogenic-Differentiated Human Bone Marrow-Derived Mesenchymal Cells Rescue Osteogenic Ability of Bone Marrow-Derived Mesenchymal Cells Impaired by Hypoxia"

_biomedicines, 2023, doi:10.3390/biomedicines11102804_

Round 1
Reviewer 1 Report
Dear editors:
It is a great honor and pleasure for me to be invited as the reviewer for this important work entitled “Extracellular Vesicles derived from osteogenic differentiated human BMSCs rescue osteogenic ability of BMSCs impaired by hypoxia”. Wang et al. investigate the molecular mechanism in vitro concerning suppressed osteogenic differentiation and simultaneously increased reactive oxygen species (ROS) production under hypoxic condition (2% O2) in human bone marrow derived mesenchymal cells (hBMSCs). This study topic is novel and advanced, attributing to Prof. Susanne Grässel’s long-term efforts and contributions in this scientific field. I have a number of comments concerning this study:
1. The hypothesized experimental design (2% O2) is a supraphysiological hypoxic condition. Animal model of Human Disease is lacking that could not be further developed as translational medicine to human beings.
2. From the perspective of a clinician involving cardiovascular research, human arterial smooth muscle cells undergoing osteogenic differentiation is reminiscent of vascular calcification and aging process. In light of this, I wonder whether profound hypoxia inhibits vascular calcification that benefits vascular health.
3. If an acronym is used in the abstract/ keywords, it must be spelled out. BMSC is the abbreviation of “bone marrow derived mesenchymal cells”, lacking in definition in the abstract (Line 22) and keywords. It should be spelled out for the first time.
4. Did anti-oxidant therapy improve the suppressed osteogenic ability of hBMSCs under supraphysiological hypoxic condition?
5. I wonder if readers could directly visualize the stages of membrane fusion using cryo-electron tomography?
6. I think some of my questions are difficult to answer. In fact, the scientific merit of the study is advanced that could be published in Biomedicines after appropriate revision.
Please see above comments.
Author Response
We are grateful for your opinion on the manuscript and your helpful comments. Thanks to your comments we were able to improve the quality of our manuscript. In the following you will find our point-by-point responses to each of the comments of yours. Changes in the manuscript are marked in yellow.
Reviewer 1
Comment: The hypothesized experimental design (2% O2) is a supraphysiological hypoxic condition. Animal model of Human Disease is lacking that could not be further developed as translational medicine to human beings.
Response: As you mentioned, direct, noninvasive in vivo measurement of oxygen tension in bone marrow faces a major technical hurdle, so animal models of human diseases are scare or lacking. However, Fischer et al. (1) showed that oxygen tension could be measured with flexible polarographic microelectrodes in the oviduct and uterus of rhesus monkeys, hamsters and rabbits, and the intrauterine oxygen tension in monkey was about 1.5%.
Comment: From the perspective of a clinician involving cardiovascular research, human arterial smooth muscle cells undergoing osteogenic differentiation is reminiscent of vascular calcification and aging process. In light of this, I wonder whether profound hypoxia inhibits vascular calcification that benefits vascular health.
Response: It is a very bright and interesting idea. According to our knowledge, different cell types prefer different oxygen tensions, for example, Markway et al. (2) showed that 2% O2 can enhance chondrogenic differentiation of human BMSC. However, some studies (3) reported that hypoxia rather increase vascular calcification than to inhibit it indicating ambivalent effects of hypoxia.
Comment: If an acronym is used in the abstract/ keywords, it must be spelled out. BMSC is the abbreviation of “bone marrow derived mesenchymal cells”, lacking in definition in the abstract (Line 22) and keywords. It should be spelled out for the first time.
Response: As suggested, the wording in the abstract has been modified appropriately.
Comment: Did anti-oxidant therapy improve the suppressed osteogenic ability of hBMSCs under supraphysiological hypoxic condition?
Response: Yes, Ma et al. (4) reported that icariin rescued the osteogenic ability of osteoblasts impaired by hypoxia via reducing production of ROS, increasing anti-oxidant SOD and ALP activity and formation of mineralized matrix.
Our own actual study showed that norm-osteo EVs are enriched with three antioxidant proteins that can rescue the osteogenic ability of naïve hBMSCs cultured under hypoxia and reduce hypoxia-induced elevation of ROS production.
Comment: I wonder if readers could directly visualize the stages of membrane fusion using cryo-electron tomography?
Response: It is a great idea and we guess that this might be possible. Unfortunately, this technique is very specialized and expensive and not commonly available. So neither our University nor other collaboration partners have a cryo-electron tomograph.
References
- B. Fischer, B. D. Bavister, Oxygen tension in the oviduct and uterus of rhesus monkeys, hamsters and rabbits %J Reproduction. 99, 673-679 (1993).
- B. D. Markway et al., Enhanced Chondrogenic Differentiation of Human Bone Marrow-Derived Mesenchymal Stem Cells in Low Oxygen Environment Micropellet Cultures. Cell Transplantation 19, 29-42 (2010).
- D. M. Csiki et al., Hypoxia-inducible factor activation promotes osteogenic transition of valve interstitial cells and accelerates aortic valve calcification in a mice model of chronic kidney disease. Frontiers in Cardiovascular Medicine 10, (2023).
- H. P. Ma et al., Icariin attenuates hypoxia-induced oxidative stress and apoptosis in osteoblasts and preserves their osteogenic differentiation potential in vitro. Cell Proliferation 47, 527-539 (2014).
Reviewer 2 Report
In the manuscript: “Extracellular Vesicles derived from osteogenic differentiated human BMSCs rescue osteogenic ability of BMSCs impaired by hypoxia”, the authors discussed the role of extracellular vesicles in reducing hypoxia production in osteogenic differentiated hBMSCs by inducing expression of anti-hypoxic/ antioxidant and pro-osteogenic genes.
Overall, this manuscript results very interesting, the authors clearly explain the rational of the study and discussed the topic point by point.
However, we would like to invite the authors to clarify some critical points:
1. Please check the check punctuation and spaces;
2. Please try to re-arrange the abstract because it is composed only of aim and results, a bit of introduction should be useful;
3. Page 2; lines 50-59, the authors said “Hypoxia is able to impair bone regeneration via reducing the differentiation capacity of bone marrow derived mesenchymal cells (BMSC) towards osteoblasts [6]. However, there are also other reports showing conflicting and inconsistent results regarding the influence of hypoxia on osteogenic differentiation of precursor cells. Wagegg et al. [7] showed that osteogenic differentiation of naïve human (h)BMSCs is enhanced under hypoxic conditions compared to normoxic conditions. They concluded that hypoxia promotes osteogenesis but suppresses adipogenesis of human MSCs in a competitive and Hypoxia‐inducible factor (HIF)-1-dependent manner. HIFs are proteins that respond to changes in oxygen concentration. HIF-1α is subjected to proteosomal degradation under normoxia, while it is stabilized under hypoxia [8]”. This concept is a little confused, are you talking about BMSCs or MSCs in general? What is the bind between osteogenesis and adipogenesis under hypoxic conditions? Please try to better explain;
4. Within the introduction, it should be useful briefly introduce in general the role of EVs in mesenchymal stem cells differentiation, please add the relative and recent, reference;
5. Within materials and methods section the authors described the isolation hBMSCs, did you check the expression of stem cells specific biomarkers before the use? Are you sure you did not use fibroblast cells? Please insert the results of cells characterization if available or eventually explain why did not do;
6. Figure 2 E-F: what is the magnification used? Please insert this information;
7. Usually the signals of western blotting are normalized with respect an housekeeping protein (e.g. tubulin, actin, GAPDH). Please, explain why did not you do;
8. Figure 3; the western blotting relative to RUNX-2 appears to saturation, maybe the results are not consistent. Could you provide acquisition?
9. Figure 5; it is complicate to read. Could you improve the resolution?
minor spelling mistakes are present
Author Response
We are grateful for your opinion on the manuscript and your helpful comments. Thanks to your comments we were able to improve the quality of our manuscript. In the following you will find our point-by-point responses to each of the comments of yours. Changes in the manuscript are marked in yellow.
Reviewer 2
Comment: Please check the check punctuation and spaces
Response: As requested, punctuation and spaces have been checked and modified appropriately.
Comment: Please try to re-arrange the abstract because it is composed only of aim and results, a bit of introduction should be useful;
Response: As requested, the introduction in the abstract has been extended appropriately.
Comment: Page 2; lines 50-59, the authors said “Hypoxia is able to impair bone regeneration via reducing the differentiation capacity of bone marrow derived mesenchymal cells (BMSC) towards osteoblasts [6]. However, there are also other reports showing conflicting and inconsistent results regarding the influence of hypoxia on osteogenic differentiation of precursor cells. Wagegg et al. [7] showed that osteogenic differentiation of naïve human (h)BMSCs is enhanced under hypoxic conditions compared to normoxic conditions. They concluded that hypoxia promotes osteogenesis but suppresses adipogenesis of human MSCs in a competitive and Hypoxia‐inducible factor (HIF)-1-dependent manner. HIFs are proteins that respond to changes in oxygen concentration. HIF-1α is subjected to proteosomal degradation under normoxia, while it is stabilized under hypoxia [8]”. This concept is a little confused, are you talking about BMSCs or MSCs in general? What is the bind between osteogenesis and adipogenesis under hypoxic conditions? Please try to better explain;
Response: Thanks for pointing this out. Firstly, we are not meaning MSCs in general but specifically bone marrow derived MSCs. As suggested, we have deleted the “….but suppresses adipogenesis of human MSCs in a competitive….” sentence to make it clearer for the reader because it is not the main purpose of our study to analyse adipogenesis of hBMSCs. In our previous study (1), we introduced the hypothesis that the cross-talk between adipose tissue and bone constitutes a homeostatic feedback loop with adipose tissue and bone regulating each other in a complex feedback system. Therefore, any adipogenic or osteogenic differentiation disorder of BMSCs may contribute to bone loss during aging, with some pathological processes are accompanied by increased bone marrow adipogenesis.
Comment: Within the introduction, it should be useful briefly introduce in general the role of EVs in mesenchymal stem cells differentiation, please add the relative and recent, reference;
Response: As suggested, the introduction has been modified appropriately (page 2, lines 81-87 and page 3, lines 103-108).
Comment: Within materials and methods section the authors described the isolation hBMSCs, did you check the expression of stem cells specific biomarkers before the use? Are you sure you did not use fibroblast cells? Please insert the results of cells characterization if available or eventually explain why did not do
Response: Yes, we do this at random. We have now provided the latest characterization of immunophenotype of hBMSCs by Flow cytometric analysis of hBMSC with specific antibodies against positive marker (CD44, CD73) and negative marker (CD19, CD34). Please see the Suppl. figure 1 and the method section 2.2 for your information.
Comment: Figure 2 E-F: what is the magnification used? Please insert this information;
Response: It is 20 x 10 magnification, which we have added to the figure legend.
Comment: Usually the signals of western blotting are normalized with respect an housekeeping protein (e.g. tubulin, actin, GAPDH). Please, explain why did not you do;
Response: Because tubulin, actin and GAPDH are located in the cytoplasm, while HIF-1α and RUNX2 are located in nucleus. We therefore have chosen the nuclear protein Histone H3 as a housekeeping protein.
Comment: Figure 3; the western blotting relative to RUNX-2 appears to saturation, maybe the results are not consistent. Could you provide acquisition?
Response: The image acquisition time we applied for figure 3 is 8 minutes (SuperSignal West Femto), because the signal of HIF-1α appeared too weak when using a shorter time of image acquisition. Please see the figure below for your information where we have stopped image acquisition after 3 minutes.
Comment: Figure 5; it is complicate to read. Could you improve the resolution?
Response: Yes, the new images with a better resolution are integrated in figure 5. However, due to the magnitude of information provided by enrichment analyses the fond cannot be increased further and will always remain a bit difficult to read.
References
- C. L. Wang et al., Effects of Extracellular Vesicles from Osteogenic Differentiated Human BMSCs on Osteogenic and Adipogenic Differentiation Capacity of Naive Human BMSCs. Cells 11, (2022).

Round 2
Reviewer 2 Report
The manuscript content and quality have been improved.
Please just check minor mistakes of spelling or writing